# Vertebrate *myosin 1d* regulates left–right organizer morphogenesis and laterality

Manush Saydmohammed[1], Hisato Yagi [1], Michael Calderon[2], Madeline J. Clark[3], Timothy Feinstein[1], Ming Sun[2], Donna B. Stolz[2], Simon C. Watkins[2], Jeffrey D. Amack[3], Cecilia W. Lo [1] & Michael Tsang [1]

Establishing left–right asymmetry is a fundamental process essential for arrangement of visceral organs during development. In vertebrates, motile cilia-driven fluid flow in the left–right organizer (LRO) is essential for initiating symmetry breaking event. Here, we report that *myosin 1d* (*myo1d*) is essential for establishing left–right asymmetry in zebrafish. Using super-resolution microscopy, we show that the zebrafish LRO, Kupffer's vesicle (KV), fails to form a spherical lumen and establish proper unidirectional flow in the absence of *myo1d*. This process requires directed vacuolar trafficking in KV epithelial cells. Interestingly, the vacuole transporting function of zebrafish Myo1d can be substituted by myosin1C derived from an ancient eukaryote, *Acanthamoeba castellanii*, where it regulates the transport of contractile vacuoles. Our findings reveal an evolutionary conserved role for an unconventional myosin in vacuole trafficking, lumen formation, and determining laterality.

[1] Department of Developmental Biology, University of Pittsburgh, 3501 5th Avenue, Pittsburgh, PA 5213, USA. [2] Department of Cell Biology, University of Pittsburgh, 3500 Terrace Street, Pittsburgh, PA 15261, USA. [3] Department of Cell and Developmental Biology, State University of New York Upstate Medical University, 750 East Adams Street, Syracuse, NY 13210, USA. Correspondence and requests for materials should be addressed to M.S. (email: mas386@pitt.edu) or to M.T. (email: tsang@pitt.edu)

The biological cues that regulate how and when left–right (LR) patterning is established during development has been a fascinating question that remains unresolved. Several animal models have been used to investigate different theories for establishment of LR asymmetry[1]. This includes intracellular chirality[2], voltage gradient flow[3], chromatid segregation[4], and motile cilia[5]. Cilia-driven fluid flow has been

reported to be essential for left–right asymmetry in *Danio rerio* (zebrafish)[6], *Xenopus laevis* (African Clawed Frog)[7], *Mus musculus* (mouse)[8], and *Oryctolagus cuniculus* (rabbit)[9]. However, a role for cilia is not universally conserved across vertebrates. In chick embryos, symmetry breaking is regulated by asymmetric cell migration around Hensen's node and does not involve cilia-based flow[10]. In addition, motile cilia are not associated with LR

asymmetries that develop in invertebrates[11,12], and for asymmetric looping of the gut and gonad in *Drosophila*[13,14]. Therefore, probing common cellular events in different model systems may reveal insights into the early mechanisms of breaking symmetry.

De novo lumen formation by cord hollowing, cavitation, and membrane repulsion has been described for tubular organs during development[15]. Lumen formation by cord hollowing begins with cellular rearrangement of non-polarized cells to establish a polarized rosette structure that dictates an apical membrane surface. Luminal growth begins with trafficking of membrane vesicles targeted to the apical surface where it fuses with plasma membrane. This process also delivers fluid at the apical surface to expand the lumen. In zebrafish, KV lumen formation is a cord hollowing process[16] and its threshold size is critical for creating a unidirectional flow and consistent laterality[6]. How the KV, a near perfect spherical structure forms at the base of the notochord is not fully established. Evidence from studies on ion channels, such as Cystic fibrosis transmembrane conductance regulator (Cftr) and $Na^{+}/K$-ATPase, as well as tight junction proteins show that they are critical for establishing KV lumen size[17–20]. However, additional mechanisms that regulate cellular remodeling and specific lumen shape is not known. Here, we reveal an unconventional myosin, *myo1d* is required for proper KV lumen formation in zebrafish. For this, vacuolar trafficking by Myo1d and expulsion of intracellular fluid from KV epithelial cells into the lumen is essential for attaining definitive spherical shape and threshold lumen size that specifies embryo laterality.

## Results

**Myo1d is required for proper LRO formation.** Prior studies involving *Drosophila* Myosin1D revealed a role for actin-based molecular motors in establishing LR asymmetry in invertebrates[13]. We wondered whether *myo1d* has an evolutionary conserved role in specifying laterality in vertebrates. Zebrafish *myo1d* is expressed early and ubiquitously, including within the tailbud region where the ciliated KV forms (white arrow, Fig. 1a). We generated *myo1d* mutations using goldy TALENs[21–23] to disrupt the motor domain and reveal its role in LR patterning (Fig. 1b). Three mutant alleles of *myo1d*, herein named *pt31a, b,* and *c* were predicted to cause frame shift and protein truncation (Fig. 1b). However, zygotic homozygous mutants (*myo1d*^pt31a/pt31a^) survived to adulthood in a Mendelian ratio suggesting that *myo1d* maternal expression was sufficient for embryonic development. We generated *myo1d*^pt31a/pt31a^ maternal-zygotic (*MZ*) mutants and found that the KV forms into either a small or dysmorphic lumen at the eight somite stage (S), approximately 11 h post fertilization (hpf) (Fig. 1c). Consistently, injection of *myo1d* translation blocking antisense morpholinos (MO) also decreased KV size (Supplementary Fig. 1a, b & Supplementary Movie 1 & 2). Next, we stained for the tight junction protein, ZO-

1 to reveal apical KV morphology. In *myo1d MZ* mutants, ZO-1 expression revealed small or dysmorphic KV shape compared to controls (Fig. 1c, and Supplementary Fig. 1c). We crossed *myo1d*^pt31a^ mutants into a transgenic line that labels membranes of KV cells[24], *Tg(dusp6:GFP-MA)*^pt21^, and confirmed that *myo1d MZ* mutants have smaller lumen (Fig. 1c, d). Dysmorphic shape of *myo1d MZ* mutants were further quantified for circularity or isoperimetric quotient, a function of perimeter and area[25]. We first analyzed KV lumen circularity in embryos during somitogenesis stages to show that as the lumen expands, almost perfect circularity was maintained (Supplementary Fig. 1d, e). In contrast, compared to WT embryos, circularity of KV were decreased in *myo1d MZ* mutants (Fig. 1c, e). These analyses indicate that *myo1d* is essential for generating a spherical KV shape and forming a threshold lumen size by 8S. A similar lumen formation defect was also observed in the otic vesicle (OV) (Fig. 1f–h), suggesting that *myo1d* is essential for the formation of these fluid-filled structures during development. We confirmed that Myo1d expression was detected in KV epithelial cell borders (Supplementary Fig. 2a), whereas Myo1d was absent in mutant KVs (Supplementary Fig. 2b). These data suggest that Myo1d plays a role in the KV epithelial cells and that the *pt31a* allele is a null mutant.

**myo1d MZ mutants display LR patterning defects.** Several reports have shown that a minimum threshold lumen size is necessary for establishing robust KV flow and LR asymmetry in zebrafish[18–20,26,27], which prompted us to investigate the mechanism of KV lumen formation. At 6 hpf, the KV is formed from a group of cells called dorsal forerunner cells (DFCs) that form at the base of the organizer[6]. These cells proliferate and undergo a mesenchymal-to-epithelial transition to form a rosette-like structure, where KV lumen forms[6]. To assess if the abnormal KV lumen phenotype in *myo1d MZ* mutants are a result of defects in DFC clustering, we analyzed *foxj1a* expression as a marker for DFCs[28]. We did not observe differences in *foxj1a* expression between *myo1d MZ* and wild-type embryos (Supplementary Fig. 3a). This indicated that DFC clustering and migration was not the cause of defective KV morphogenesis in the *myo1d MZ* mutants. One of the earliest transcriptional responses to proper KV function is asymmetric expression of the TGFβ antagonist, *dand5* (also known as *charon*)[29]. In *myo1d MZ* mutants, the asymmetric *dand5* expression on the right- versus left-side KV were reduced when compared to WT embryos at 8S stage indicative of a failure of proper KV function (Fig. 2a, b). We next assessed the expression of *spaw*, a *Nodal* gene that marks establishment of definitive left-sided patterning at 18S. In *myo1d MZ* mutants normal left-sided *spaw* expression was disrupted such that embryos presented bilateral and even right-sided expression (Fig. 2c, d). We confirmed that the bilateral *spaw* was not a consequence of midline defects as both *T-box transcription*

**Fig. 1** *myo1d* is required for Kupffer's vesicle morphogenesis. **a** *myo1d* is ubiquitously expression. White arrow demarcates the region where KV is formed. **b** TALEN mediated genome editing of *myo1d*. TALENs were designed to target *myo1d* exon 2 (red arrow) encoding the myosin motor domain (blue). Calmodulin binding IQ motif (green) and Tail Homology 1 (TH1, red) domains are also indicated. Three F1 alleles were isolated, *pt31a-c*, with predicted amino acid sequence indicated. **c–e** *myo1d MZ* mutants showed KV lumen formation defects. Representative DIC images showing KV of *myo1d MZ* mutants were either smaller and dysmorphic lumens (**c**). ZO-1 staining at 8S confirmed that the apical surface of KVs were smaller or dysmorphic in *myo1d* mutants when compared to controls. *myo1d MZ;Tg(dusp6:GFP-MA)* injected with *H2BmCherry* mRNA showing dysmorphic KV epithelial cell morphology and decreased lumen area, when compared to controls. Lumen marked by dotted circular lines were quantified for area (WT: $n = 44$, *myo1d MZ*: $n = 46$) (**d**) and circularity (WT: $n = 20$, *myo1d MZ*: $n = 21$) (**e**). **f–h** Dysmorphic otic vesicle (OV) observed in *myo1d MZ* embryos. *Tg(dusp6:GFP-MA)* and *myo1d MZ;Tg(dusp6:GFP-MA)* embryos (**f**) or stained with aPKC (green) and acetylated tubulin (red) (**g**) at 24 hpf show smaller OVs and quantified (WT: $n = 8$, *myo1d MZ*: $n = 12$) (**h**). KV lumen was considered small when it was below threshold lumen size (1000 μm$^2$). Unpaired $t$ test was used with, ***$p < 0.001$, ****$p < 0.0001$ indicated. Data shown in the graphs are mean ± SEM. Scale bar, 50 μm

*factor Ta* (*tbxta*, also known as *ntla*) and *myogenic differentiation 1* (*myod1*) expression at 10S and 17S were unaffected in *myo1d MZ* mutants (Supplementary Fig. 3b). Consistently, we detected increased frequency of laterality defects in *myo1d MZ* mutants (Fig. 2c, d) and in *myo1d* morphants (Supplementary Fig. 1f).

These results suggest that laterality defects we observed in *myo1d* mutants are not primarily due to midline formation defects, but a result of proper KV formation.

Since motile cilia in the KV are required for a unidirectional flow and proper laterality[30], we determined if loss of *myo1d*

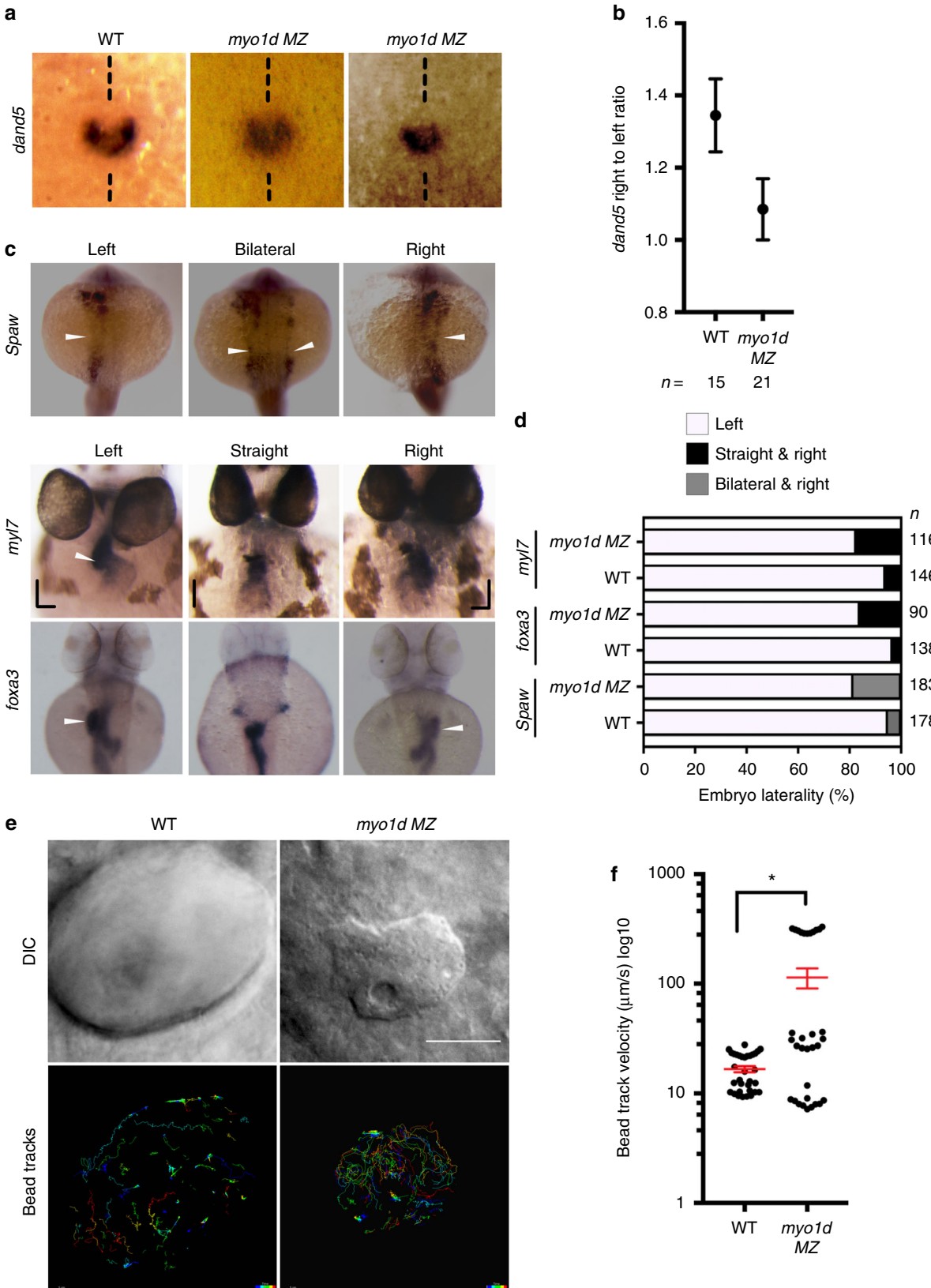

affected ciliogenesis. Acetylated tubulin staining showed that cilia length was normal (Supplementary Fig. 4a, b), but cilia number per KV were less in *myo1d MZ* mutants (Supplementary Fig. 4c). As KV epithelial cells are monociliated[6], we counted KV nuclei bordering the lumen and found fewer cells in *myo1d MZ* embryos that accounted for the decrease in cilia number (Supplementary Fig. 4d, e). Given that cilia length and number were minimally affected in *myo1d MZ* mutants, we next determined if disruption in KV fluid flow could explain for the observed laterality defects. To measure fluid flow, we tracked the movement of FITC fluorescent microbeads injected into the KV of WT and *myo1d MZ* mutants. We noted in *myo1d MZ* mutants, unidirectional KV fluid flow was lost compared to the control embryos (Fig. 2e, f, and Supplementary Movie 3 & 4). Together, these experiments revealed that in the absence of Myo1d, the process of forming proper KV shape and size is disrupted that results in failure of unidirectional fluid flow and ultimately disrupting LR patterning.

**Vacuolar trafficking in KV lumen is disrupted in *myo1d* mutant**. KV lumen formation is a rapid and dynamic process during somitogenesis that spans approximately 3-h period (1S–8S, 10–13 hpf) (Supplementary Movie 1)[6]. Previous studies described how a symmetric KV rosette undergoes extensive cellular remodeling to become an asymmetric KV so that anterior cells retain columnar shape whereas posterior cells attain squamous or cuboidal shape[31]. Depletion of Rock2b or pharmacological inhibition of non-muscle myosin II altered anterior–posterior (AP) cell shape changes in the KV[21], suggesting that actomyosin activity is important for generating AP asymmetry. Additional fluid filling mechanisms also contribute to this process as chemical inhibition of ion channels, knockdown of cell junction proteins, and mutations in Cftr provide insights into their role in proper KV formation[17–20]. Thus, we reasoned that *myo1d* could contribute to the AP asymmetry and lumen expansion through a fluid filling mechanism. In *Tg(dusp6:GFP-MA)* embryos, we observed vacuole-like structures in the KV cells that were designated as such by virtue of their larger size when compared to endocytic or ER-derived vesicles[32] (Fig. 3a, and Supplementary Movie 5 & 6). We hypothesized that vacuoles trafficking to the apical surface could be the basis of how the KV lumen expands as these structures provide both membrane and fluid to the epithelium. To quantify this, we counted the number of vacuoles in the KV cells at the beginning and the end of lumen expansion phase (from 3S to 8S). At 3S, vacuoles appeared near the plasma membrane of every KV epithelial rosette cell (Fig. 3a). By 8S, the number of vacuoles was significantly less in posterior cells (Fig. 3a, b). In *myo1d MZ* mutants and *myo1d* AUG morphants, we observed an increased number of vacuoles and persistence of columnar shape in the posterior KV epithelial cells at 8S (Fig. 3a, c). Consistently, transmission electron microscopic images revealed an abundance of vacuole-like structures in the *myo1d MZ* KVs (Fig. 3d). Moreover, Myo1d was co-localized with the vacuoles in KV cells (white arrow, Supplementary Fig. 2a), suggesting Myo1d is involved in trafficking of these vacuoles. Together, these experiments suggest that vacuole clearance contributes to AP cell shape changes in the KV. Next, we

used super-resolution imaging to precisely track vacuolar movement from 2S to 3S. The epithelial cells lining KV showed directed vacuolar trafficking toward the rosette apex (Fig. 3e, and Supplementary Movie 7). This vacuolar movement is dynamic such that smaller vacuoles can fuse together as they reach the KV apical membrane (Fig. 3e, & Supplementary Movie 7). On the contrary, *myo1d MZ* embryos revealed a dispersed, largely cytoplasmic distribution of vacuoles and their movements were misdirected (Fig. 3e). Moreover, the speed of vacuolar movement was decreased in *myo1d MZ* mutants (Fig. 3e, f, and Supplementary Movie 8). Continuous vacuole fusion events (white arrow Fig. 3g) resulted in larger vacuoles (compare vacuoles marked with white spots in Fig. 3g, and Supplementary Movie 7 & 8). Further, larger vacuoles were predominant in KV epithelial cells of *myo1d MZ* mutants (Fig. 3h, and Supplementary Movie 9 & 10). Thus, *myo1d* is required to deliver fluid filled vacuoles to the KV lumen apex. These observations establish the role of *myo1d* in directed vacuolar movement as a mechanism for KV lumen expansion.

**Conservation of Class 1 myosin function in KV morphogenesis**. In ancient unicellular eukaryotes, myosin-I is involved in transporting water through contractile vacuoles (CV), which is essential for attaining amoeboid cell shape and regulating directed cell motility[33,34]. Loss of *Acanthamoeba castellanii* myosin-IC activity by inhibitory antibodies resulted in CV accumulation in the cell that ultimately leads to cell rupture[33]. We reasoned that a similar process is occurring in the KV epithelial cells in *myo1d MZ* mutants. We found a significantly higher number of fragmented nuclei in *myo1d MZ* embryos compared to wild type at 1S and 6S stage (Fig. 4a, b), suggesting that accumulation of vacuoles may cause KV epithelial cells to burst that may also account for the lower cell number (Supplementary Fig. 4d, e). All unconventional Myosin I class proteins contain a Myosin motor at the N-terminus, and a Myosin Tail-Homology 1 (TH1) domain at the C-terminus[35]. Other structural domains that are shared within this class of proteins include calmodulin binding IQ motifs and SH3 domains (Supplementary Fig. 5a)[35]. We used Phyre2 to predict the motor and TH1 protein structures as conservation between zebrafish Myo1d and amoeba myosin-1C were lower than the vertebrate Myosins (Supplementary Fig. 5a)[36]. Both the motor and TH1 domains (Supplementary Fig. 5b) from these species show similar predicted structures suggesting that they could serve similar functions. Hence, we postulated that the *Amoeba* myosin-1C could compensate for the loss of *myo1d* in zebrafish. Overexpression of *Acanthamoeba* myosin-IC artificially expanded KV lumen in wild-type embryos (Supplementary Fig. 5c, d), indicating that myosin-IC derived from *Amoeba* is functional in zebrafish KV cells. Moreover, injection of *Acanthamoeba* myosin-IC mRNA into *myo1d MZ* embryos rescued KV lumen and AP cell morphology defects as efficiently as a phylogenetically closer version of *Myosin 1d* derived from rat (Fig. 4c, d). Rescue of circularity, reduction of vacuole volume, and laterality from ectopic expression of rat Myo1d was also observed in Myo1d-deficient embryos (Fig. 4e, f and Supplementary Fig. 1g). These results suggest that vacuole delivery is

---

Fig. 2 *myo1d* is necessary for KV function and establishing laterality. **a, b** Relative abundance of *dand5* expression on the right vs left were decreased in *myo1d MZ* mutants compared to WT embryos at 8S. **c, d** *myo1d MZ* mutants displayed laterality defects. Heart and gut looping scored by *myl7* and *foxa3* expression at 72 hpf respectively, spaw expression at 17S (white arrowhead marks expression). Laterality defects were scored in **d**. **e, f** *myo1d MZ* embryos display altered KV flow as analyzed by tracking 0.2 μ FITC fluorescent microbeads injected into KV lumen. DIC images showing KV morphology of control and *myo1d MZ* mutants (**e**). Individual beads were mapped showing unidirectional circular movement were lost in *myo1d MZ* mutants compared to wild-type embryos. Individual bead movement was shown as temporal color-coded tracks. Graph comparing bead track velocity (mean ± SEM) in control ($n = 33$ tracked beads from three embryos) and *myo1d MZ* mutant embryos ($n = 33$ tracked beads from three embryos) (**f**). Unpaired $t$ test indicate statistical significance * $p < 0.05$. Scale bar, 50 μm

important for lumen expansion and *myo1d* is essential for this process. All Myosin I molecules contain a TH1 domain that binds to lipid membranes[35,37]. To further understand if TH1 domain in

Myo1d is required for KV lumen formation, we injected rat Myo1d lacking the TH1 domain (*Myo1dΔTH1*) into *myo1d MZ* mutants and observed a failure of KV lumen size rescue (Fig. 4c,

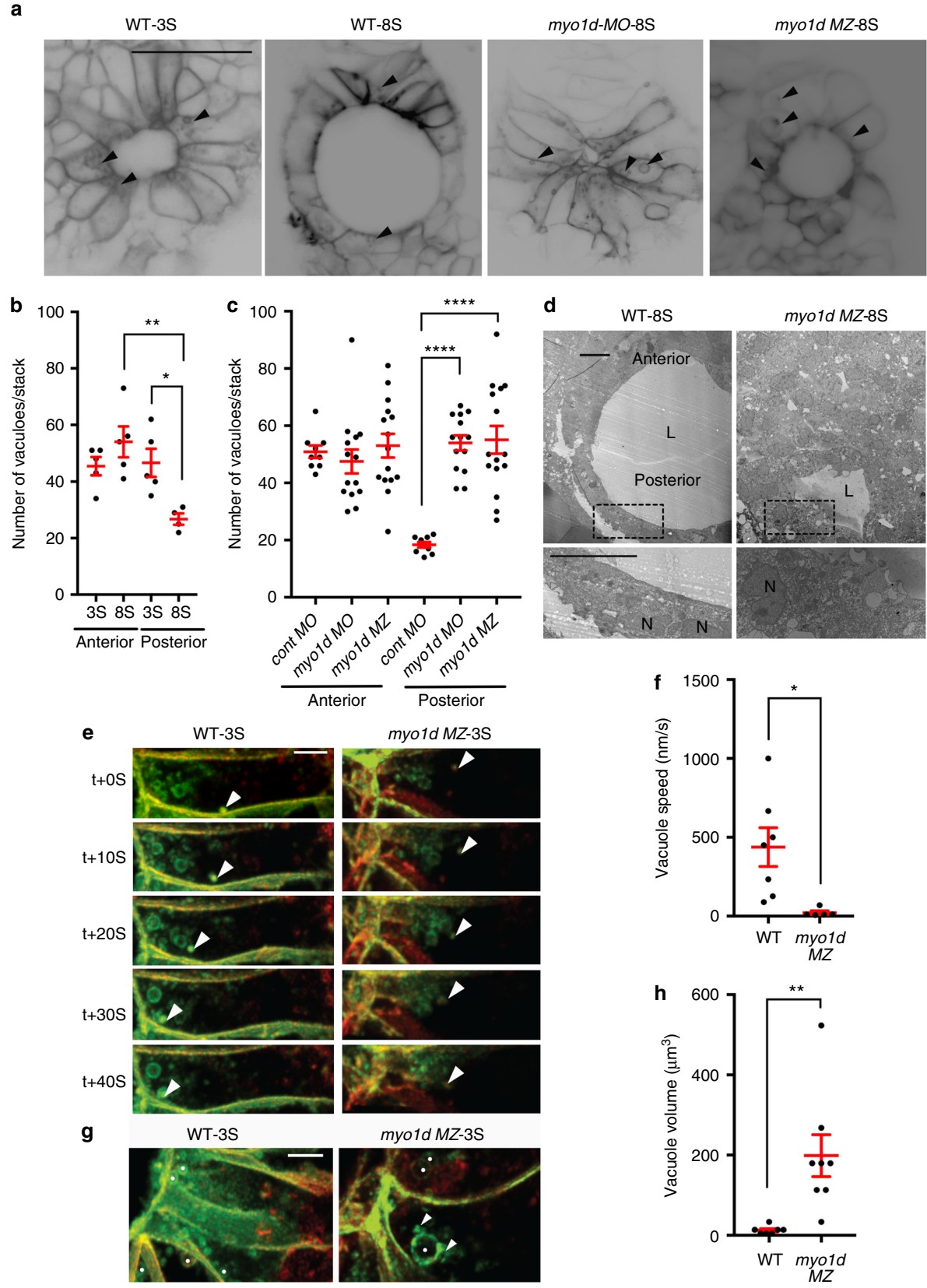

**Fig. 3** *myo1d* regulate directed vacuolar movement in KV cells. **a–c** Anterior and posterior KV epithelial cells contain numerous vacuoles (black arrow) at 3S ($n = 5$ embryos) but decrease in the posterior KV cells at 8S ($n = 5$ embryos) in *Tg(dusp6:GFP-MA)* embryos (scale bar, 50 μm). Vacuole number in posterior KV cells were increased in *myo1d* MO injected ($n = 14$) or in *myo1d MZ;Tg(dusp6:GFP-MA)* mutants ($n = 15$) when compared to control MO injected embryos ($n = 9$). Numeration of vacuole number per KV stack are shown in **b** and **c**. **d** TEM showing presence of vacuoles (black arrow) in posterior KV apex in *myo1d MZ* but not in WT embryos at 8S. L lumen, N nucleus, ($n = 4$) (scale bar, 10 μm). **e–h** 4D STED time-lapse still images of a representative KV epithelial cell. *Tg(dusp6:GFP-MA)* KV injected with *H2BmCherry mRNA* showing directed vacuole migration toward apical surface (white arrow), appeared to fuse with luminal surface. Vacuole movement toward KV apex in the *myo1d MZ;Tg(dusp6:GFP-MA)* mutant were decreased (**e**). Average vacuole movement speed in KV epithelial cells in *myo1d MZ* mutants ($n = 5$ vacuoles from three embryos) were decreased compared to WT ($n = 7$ vacuoles from three embryos) (**f**). In *myo1d MZ;Tg(dusp6:GFP-MA)* embryos ($n = 7$ vacuoles from three embryos), vacuole fusion (shown as white arrow) resulted in increased vacuole size when compared to *Tg(dusp6:GFP-MA)* control ($n = 8$ vacuoles from three embryos) (**g**). Vacuoles denoted by white spots. Average vacuole volume was increased in KV epithelial cells (**h**). Statistical analysis by one-way ANOVA and post hoc analysis with Turkey's multiple range tests or unpaired *t* test. * $p < 0.05$, **$p < 0.01$, ***$p < 0.005$, ****$p < 0.0001$ represents a statistical difference. Data shown are mean ± SEM. Scale bar, 5 μm

d). These results indicate that *myo1d* is essential component for KV morphogenesis and function.

**Cftr functions independently of Myo1d in the KV**. Cftr, which is localized in KV apical membranes, was shown to regulate water transport into the epithelial lumen[38]. Consistently, zebrafish Cftr mutants exhibited impaired KV lumen formation[19], while activating Cftr with Forskolin and IBMX (FIBMX) expanded KV lumen size[20]. Myo1a is required for Cftr brush border membrane trafficking and ion transport in the mouse intestine epithelium[39]. So, we wondered whether Cftr in KV apical surface is regulated through *myo1d* in zebrafish. This mechanism could also account for the smaller KV lumen in *myo1d MZ* mutants. For this, we quantified Cftr trafficking to the KV apical surface using the *TgBAC(Cftr-GFP)* embryos, where Cftr is fused to GFP[19]. We observed normal Cftr localization in the KV apical membrane after Myo1d depletion (Fig. 5a, b). Interestingly, FIBMX treatment in *myo1d MZ* embryos showed expansion of KV lumen, implicating Cftr mediated lumen expansion was still functional in the absence of Myo1d (Fig. 5c, d). However, FIBMX treatment in *myo1d MZ* mutants failed to restore circular KV shape and cardiac looping (Fig. 5e). We next determined if in the absence of Myo1d, suppressing $Na^+/K^+$-ATPases would have an additive effect on lumen size. The ion gradient driving Cftr activity is generated by $Na^+/K^+$-ATPases in the KV[19,20]. Embryos treated with ouabain, a potent specific $Na^+/K^+$- ATPase inhibitor, blocks Cftr function and decreases KV lumen size[19]. Ouabain treatment decreased lumen volume in control and to a lesser extent in *myo1d MZ* mutants embryos (Fig. 5c–e). Similarly, ouabain treatment increased incidence of cardiac looping defects in wild-type embryos (Fig. 5e and Supplementary Fig. 6). This is consistent with a previous observation that reported increasing laterality defects with a high dose of ouabain treatment[40]. However, ouabain treatment on *myo1d MZ* mutants had no additive effect on laterality defects (Fig. 5e). Thus, proper KV lumen formation with a spherical shape and threshold volume requires multiple modes of fluid filling mechanisms that is coordinated to expand KV lumen within a short developmental time.

## Discussion

How does Myo1d regulates laterality in different model systems? In *Drosophila*, tissue specific and temporal expression of myosin1D in the hindgut was sufficient to drive intrinsic chirality at the cellular level that generates a consistent gut and gonad looping pattern[11]. A mechanical model suggests that anchored myosin motors walking along actin drives the filaments to turn in leftward circles[41]. With no LRO in invertebrates, these circular forces drive specific organ looping morphogenesis[11,12]. On the contrary, in zebrafish, *myo1d* provides the forces required for vacuole delivery and generation of AP cell shape rearrangement of KV. In addition, vacuole

delivery by Myo1d can also provide plasma membrane for luminal surface that is required for cell shape morphology. This lack of plasma membrane delivery to the lumen could explain how definitive LRO shape fails to form in *myo1d MZ* mutants. Irregularity in KV shape likely contributes to the failure of unidirectional flow in the *myo1d MZ* mutants and increase the frequency of laterality defects. Also, during KV lumen expansion phase, AP cell shape arrangement is critical for asymmetric distribution of cilia[31]. Thus, defective epithelial cell arrangement in *myo1d MZ* mutant KVs can affect AP cilia arrangement and flow. While our manuscript was in review, two studies report that Myo1d is required for proper cilia orientation in the LRO through interaction with the planar cell polarity pathway as the reason for laterality defects[42,43]. This ciliary orientation defect contributes to disruption of unidirectional flow in the LRO (Fig. 2e, f)[42]. Our studies propose that Myo1d is required for proper KV morphogenesis and lumen formation. We reason that in the absence of this process, the proper orientation of motile cilia would likely be affected and supports these recent findings. Taken together, our results indicate that myosin motor protein found in primitive eukaryotes that is critical for LRO morphogenesis and determining LR asymmetry in vertebrates.

Although KV cell volume changes occur in concert with KV remodeling processes during lumen formation, multiple mechanisms may be involved in coordinating cell shape and threshold size of lumen by 8S stage. In support of this hypothesis, inhibiting Rock2b function or non-muscle myosin II activity had no effect on KV lumen expansion but prevented cell shape changes during KV remodeling[21]. On the other hand, Cftr, $Na^+/K^+$-ATPase ion channels and tight junction proteins regulate KV epithelial cell shape, and cell volume that impinge on proper KV lumen expansion[17–20]. In addition to these mechanisms, we propose that *myo1d*-mediated vacuolar transport is required for regulating KV epithelial cells, and lumen shape to attain a threshold volume in a limited developmental timeframe (Fig. 6).

During lumen expansion, it appears that KV fluid is primarily derived from posterior epithelial cells that decrease cell volume concomitant with lumen expansion[17]. It is remarkable that this process is akin to the water expulsion mechanism found in protozoans that traffic water and other fluid filled contractile vacuoles to the plasma membrane[33]. Similar asymmetrical fluid loss and epithelial thinning process were observed during zebrafish OV development where actomyosin interaction provides the forces necessary to expand OV lumen[44]. Consistently, this remodeling process creates new luminal space and cause a net redistribution of fluid from epithelial cells into the lumen, highlighting the role of intra epithelial fluid in lumen expansion[44]. We also observed lumen formation defects in the OV of *myo1d MZ* mutants, suggesting that *myo1d* mediated vacuole trafficking could be a common mechanism for determining

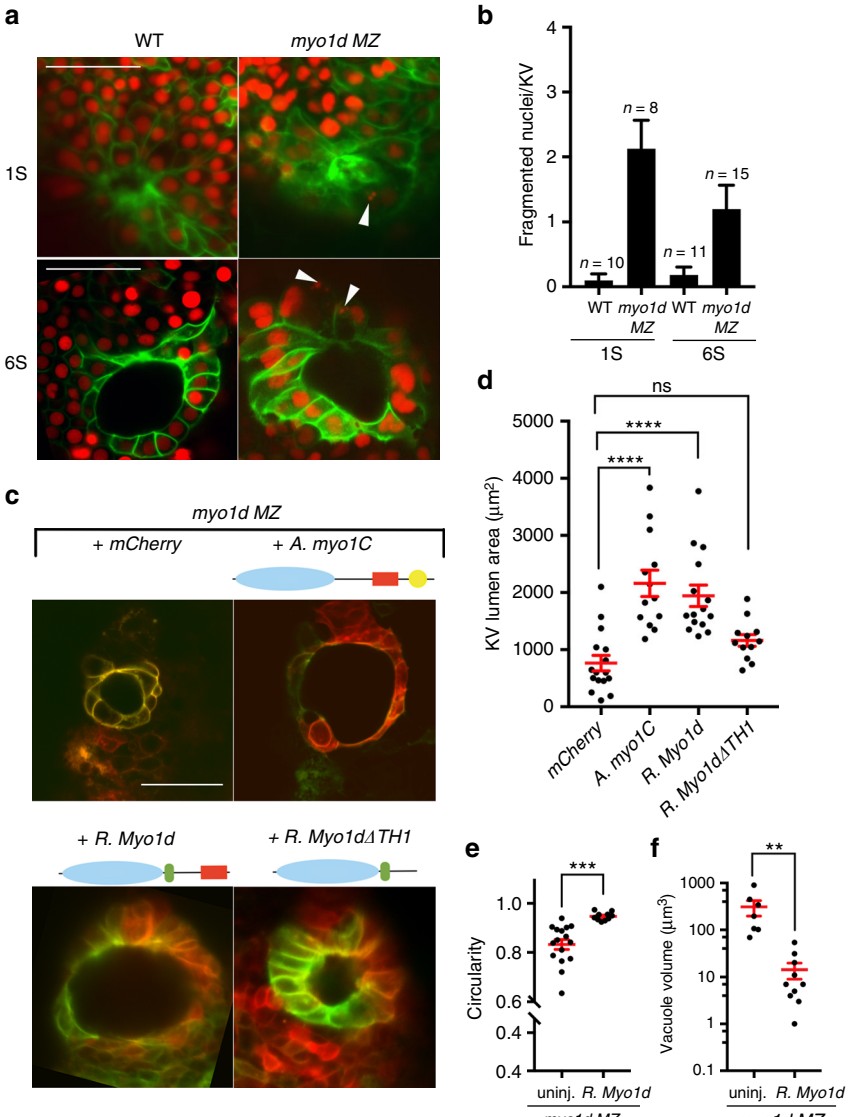

**Fig. 4** Conservation of Myosin I protein function. **a, b** Loss of *myo1d* results in increased nuclear fragmentation. *Tg(dusp6:GFP-MA)* or *myo1d MZ; Tg(dusp6: GFP-MA)* embryos injected with *H2BmCherry* mRNA exhibit fragmented nuclei in KV epithelial cells and quantified as fragmented nuclei per KV sample. Fragmented nuclei are represented as absolute numbers per KV Z-section. **c–f** Overexpression of amoeba *myo1C* (*n* = 13) or rat *Myo1d* (*n* = 15) were sufficient to rescue lumen area in *myo1d MZ* embryos compared to uninjected controls (*n* = 16) (**d**), circularity (*n* = 16 *myo1d MZ*, *n* = 10 rat *Myo1d* injected *MZ* embryos) (**e**), and reduced vacuole size (*n* = 9 *myo1d MZ*, *n* = 10 rat *Myo1d* injected *MZ* embryos) (**f**) in *myo1d MZ* mutants. However, rat Myo1d lacking the TH1 domain (*myo1dΔTH1*) (*n* = 12) failed to restore lumen size (**d**). Unpaired *t* test or one-way ANOVA with post hoc Turkey's multiple range tests were conducted with, **\*\*p* < 0.01, \*\*\*p* < 0.005, \*\*\*\*p* < 0.0001 represents statistical significance. ns not significant. Data shown are mean ± SEM

lumen shape and size during formation of fluid-filled structures in development.

## Methods

**Zebrafish handling and maintenance**. All the experiments using zebrafish was carried out with prior review and approval by the University of Pittsburgh Institutional Animal Care and Use Committee. Transgenic zebrafish lines used in work were: *Tg(dusp6:GFP-MA)[pt21]*, AB\*, *myo1d* TALEN mutant lines generated for this work were labeled: *pt31a, pt31b,* and *pt31c* alleles.

**Morpholino injections and rescue experiments**. Antisense Morpholino (MOs) were obtained from Gene-tools LLC. MOs were designed against translation initiation codon (5′-CCAAACTTTCGTGTTCTGC**CAT**AAT-3′) of zebrafish *myo1d*. MOs were injected into embryos at 2-cell stage as previously reported[45]. To conduct rescue experiments, full length rat MYO1D was PCR amplified and cloned into pCS2+ vector. Rat Myo1D TH1 deletion construct (*Myo1d ΔTH1*) was generated by primers forward (5′-gccatgtagatgTTGAAAGGTCAAAGGGCAGAC-3′) reverse (5′-tgcaacctttgcCCTGACCTGGGGAAGGTC-3′) designed to anneal full length rat MYO1D construct. Amplicons were generated and Kinase-Ligase-DpnI

(KLD) treated using Q5 site-directed mutagenesis kit (NEB#E0554S). Rat *Myo1D ΔTH1* were designed to insert a STOP codon at the start site of TH1 domain. *Acanthamoeba* sp myosin-1C[33] was cloned into pCS2+ vector between Nco1 and Xba1 restriction site. Linearized pCS2+ vector was used to generate mRNA using SP6 mMessage mMachine (Ambion: AM1340) and purified using Roche mini quick spin RNA columns (Roche: 11814427001, USA).

**Design of TALEN DNA binding domains**. The software developed by the Bogdanove laboratory (https://boglab.plp.iastate.edu/node/add/talen) was used to find best DNA binding sites to target *myo1d* exon2 as described[21]. TALEN-binding sites were 18 (Random variable repeats) RVDs (NI HD HD NH NG NH HD HD NI NG NH NI NI HD NI NG NG NG) on the left-TALEN and 19 RVDs (HD NG NH HD HD NG NG NG NH NG NI HD NG NH HD NG HD NI NI) on the right-TALEN with 15 bases of spacer.

**Construction of goldy TALENs and mutant generation**. TALEN assembly of the RVD-containing repeats was conducted using the Golden Gate approach[31]. Once assembled, the RVDs were cloned into a destination vector, pT3TS-GoldyTALEN. After sequence confirming RVDs were in frame with destination vector, linearized

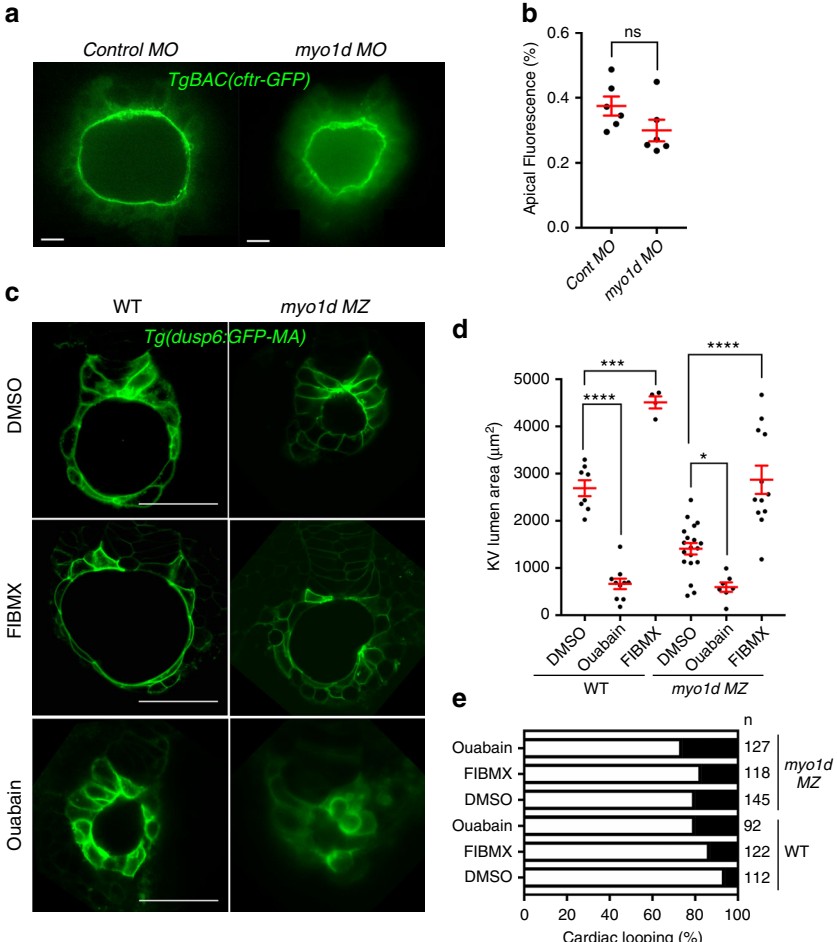

**Fig. 5** Modulating Cftr activity restores lumen size in *myo1d MZ* mutant embryos. **a, b** Knockdown of *myo1d* in *TgBAC(Cftr-GFP)* embryos did not affect CFTR apical GFP localization ($n = 6$ control MO, $n = 6$ *myo1d* MO). **c–e** Activation of Cftr function in the absence of *myo1d MZ* mutant expanded lumen area. Representative Z-stack image showing that Forskolin and IBMX (FIBMX) increased KV lumen in WT and *myo1d MZ* embryos (**c**). Treating with ouabain, an inhibitor of Na$^+$/K$^+$ ATPase decreased the KV lumen area in WT and *myo1d* mutant embryos. Numeration of lumen area with treatments are shown in (WT-DMSO: $n = 8$, WT-ouabain: $n = 10$, WT-FIBMX: $n = 4$, *myo1d MZ*-DMSO): $n = 19$, *myo1d MZ*-ouabain: $n = 7$, *myo1d MZ*-FIBMX: $n = 12$) (**d**). Cardiac looping defects were increased in WT embryos treated with FIBMX or ouabain but did not increase in *myo1d MZ* mutants (**e**). Unpaired *t* test and one-way ANOVA were conducted. *$p < 0.05$, ***$p < 0.001$, ****$p < 0.0001$ represents statistical significance. ns not significant. Data shown are mean ± SEM. L lumen, scale bar = 15 μm (**a**), 50 μm (**c**)

plasmids were used to generate mRNAs using SP6 mMessage mMachine kit (Ambion: AM1340) and purified using Roche mini quick spin RNA columns (Roche: 11814427001, USA). Each mRNA TALEN pairs were co-injected into 1-cell stage embryos and raised to adults. Somatic and heritable TALEN induced mutations were evaluated by using forward (*myo1d-F1*: 5′-TTGCTGCAGGTTTGAAAAGGGTC GTA-3′) and reverse (*myo1d-R1*: 5′-CAGTCTACCTGAGATGACAATGCACG-3′) using primers and One Taq DNA polymerase (NEB: MO0480S). PCR was performed using following cycling conditions: initial denaturation 4 min 95 °C, 35 cycles (30S 95 °C, 30S 58 °C, 30S 68 °C) and final extension 5 min 68 °C. A 226 bp PCR amplicon generated from 48 hpf embryos were used to detect disruption of AatII restriction site in the *myo1d* target site by restriction fragment length polymorphisms. Founders harboring mutant alleles were outcrossed to generate F1 population and sequence verified. Positive F1 heterozygotes were in-crossed to generate homozygous zygotic (Z) mutants. Maternal zygotic (*MZ*) mutants are generated by in-crossing homozygous parents.

**Whole-mount in situ hybridization**. Total RNA was isolated from zebrafish embryos at 24 hpf. Total RNA (1 µg) was reverse transcribed with Superscript II Reverse Transcriptase (Invitrogen) and amplified with the primers: 5′-AACGTTCCTCCTTGCCCTGTAATC-3′ (forward) and 5′-TCTATAATGTGA CCGGAGTGAGCA-3′ (reverse). PCR products of *myo1d* mRNA and inserted into PCRII-TOPO vector (Thermo Fisher Scientific: K465001). Sequence confirmed clones were used to make *myo1d* riboprobes. Following riboprobes prepared using the cDNA constructs for *myl*[46], *spaw*[47] *foxa3*[48], *foxj1a*[28], *dand5*[49], *ntla*[40], and *myod1*[40] with digoxygenin RNA labelling kit (Roche DIG RNA Labeling Kit: 11175025910). In situ RNA hybridizations in whole-mount

zebrafish embryo were performed at desired stages using standard zebrafish protocols[50]. For *dand5* quantification, right to left ratio for 8S stage embryos in WT and *myo1d MZ* were calculated using ImageJ. *dand5* insitu expression domains on either side of embryonic midline were measured after fitting a region of interest. Further, a ratio of right- vs left-sided expression region were derived to asses if KV function is compromised. This protocol was adapted from ref.[51] to describe KV function.

**Immunohistochemistry and microscopy**. Primary antibodies used for this study: acetylated tubulin (Sigma T7451: 1:500), Myosin 1d (Abcam ab70204: 1:400), and atypical PKC (Santa Cruz sc-216: 1:200). Secondary antibodies were anti mouse-Alexa 594 (Invitrogen A11005: 1:500) and rabbit-Alexa 488 (Invitrogen A11008: 1:500). Embryos were fixed in 4% Paraformaldehyde (PFA) overnight and then dechorionated in 1× PBS. Embryos were permeabilized in 1× PBS with 0.5% Triton X-100 and treated with blocking solution containing 1× PBS, 10% sheep serum, 1% DMSO, and 0.1% Triton X for an hour. Primary antibodies were diluted in fresh blocking buffer and incubated with embryos for at 4 °C overnight and then washed in 1× PBS with 0.5% Triton X-100 (3× washes, 30 min each) at room temperature.

**KV imaging and analysis**. *Tg(dusp6:GFP-MA)*[pt21] transgenic embryos at 2–3S were maintained at 28.5 °C and mounted on a transparent (MatTek: P35G-1.0-20-C) glass bottom culture dish in 1.5% low melting agarose. For Cftr expression analysis, KV was imaged in live *TgBAC(Cftr-GFP)* embryos using a Perkin-Elmer UltraVIEW Vox spinning disk confocal microscope with ×40 water immersion objective. Imaris (Bitplane) and Fiji (ImageJ NIH, Bethesda, MD) software was used to measure raw integrated density of the total KV area or the apical

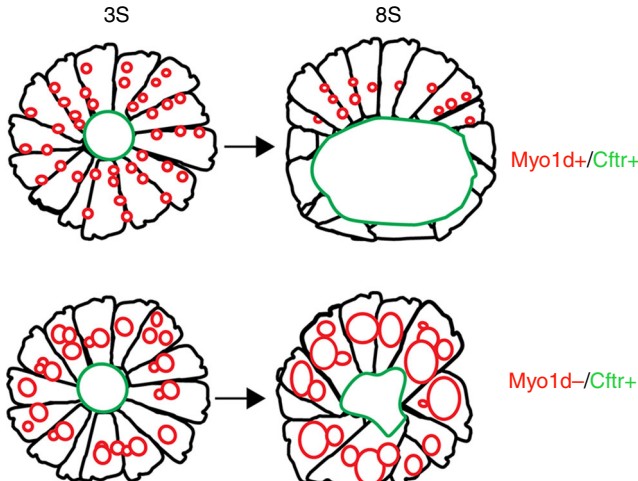

**Fig. 6** *myo1d* functions to regulate KV lumen expansion, a key event in establishing proper laterality in zebrafish. A model depicting the role of Myo1d and Cftr in KV lumen expansion. When Myo1d mediated vacuole delivery and Cftr channel are active, proper AP cell morphology and threshold lumen expansion with a definitive spherical shape are established by 8S. In the absence of Myo1d, despite CFTR channel being present, KV lumen fails to generate a spherical shape and proper size. This is due to the failure of vacuole trafficking towards apical surface of the KV

membrane region. The apical signal was then divided by the total signal to determine the apical intensity as a percentage of total fluorescence to control for differences in KV size or transgene expression levels. For four-dimensional (4D) time-lapse imaging, we used stimulation emission depletion (STED) imaging with Leica 93 × 1.3 NA motorized correction collar objective. Z stacks with 200 nm step size imaged for 10 min with 8 s intervals and were deconvolved with Huygens Professional version 17.04 (Scientific Volume Imaging, The Netherlands) in order to improve signal to noise ratio. All the measurements on speed and volume of vacuole was done manually from 4D STED images using Fiji. Vacuole volume measurements from confocal and STED images were derived from diameter, so only spherical structures were considered for measurement. To measure circularity, the largest Z stack from KV images were used to measure area and perimeter and calculate the circularity values using the formula: $f = \frac{4\pi A}{p^2}$ (see ref.[25] for a more detailed description of this method to calculate circularity).

**KV flow analysis**. Embryos were removed from chorions mounted in 1.5% low melt agarose in system water in Mattek glass bottom Petri dishes. FITC fluorescent beads (0.2 μm) (Polysciences) were pressure injected into KV at the eight-somite stage. Intact embryos after injection were remounted and imaged on a Nikon (Melville NY) Tie microscope with a high NA (1.15) ×40 LWD water objective[52]. Movies were made after collecting 2000 frames at 100 frames/s with a Photometrics 95B back-thinned sCMOS camera. Particle tracking was performed using Imaris (Bitplane) software.

**Transmission electron microscopy**. In total, 8S embryo samples were fixed overnight using cold 2.5% glutaraldehyde in 0.01 M PBS. Fixed samples were washed 3× in PBS then post fixed in aqueous 1% $OsO_4$, 1% $K_3Fe(CN)_6$ for 1 h. Following three PBS washes, the pellet was dehydrated through a graded series of 30–100% ethanol, 100% propylene oxide then infiltrated in 1:1 mixture of propylene oxide: Polybed 812 epoxy resin (Polysciences, Warrington, PA) for 1 h. After several changes of 100% resin over 24 h, pellet was embedded in molds for cross-sectioning embryos, cured at 37 °C overnight, followed by additional hardening at 65 °C for two more days. Ultrathin (70 nm) sections were collected on 200 mesh copper grids, stained using a Leica EM AC20 automatic grid staining machine with 2% aqueous uranyl acetate for 45 min, followed by Reynold's lead citrate for 7 min. Sections were imaged using a JEOL JEM 1011 transmission electron microscope (Peabody, MA) at 80 kV fitted with a side mount AMT digital camera (Advanced Microscopy Techniques, Danvers, MA).

**Pharmacological treatment**. Stock solutions of 3-Isobutyl-1-methylxanthine, 100 mM IBMX (Sigma# 410957), 10 mM Forskolin (Sigma# F3917), and 10 mM Ouabain (Cat #O 0200000) were prepared in DMSO. Embryos were treated with a working concentration of 10 μM Forskolin and 40 μM IBMX or 1 and 5 μM Ouabain in E3 egg water from bud stage until 8S.

**Statistical analysis**. Statistical significance using unpaired *t* test or one-way ANOVA and post hoc analysis using Turkey's multiple range test by Graphpad Prism (Graph pad, La Jolla, CA, USA).

**Data availability**. All data generated or analyzed during this study are included in this published article (and its Supplementary information files). In addition, data from this study are available from the corresponding authors on reasonable request.

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

## Acknowledgements

We thank E. Korn (National Institute of Health, MD) for sharing *Acanthamoeba* sp. myosin-IC plasmid construct, S. Wolfe and N. Lawson (University of Massachusetts, MA) for their suggestions on *myo1d* targeting strategies, C. Stuckenholz and L. Davidson (Department of Bio Engineering) for sharing pCS2 CAAX and H2B: mCherry plasmid construct, and K. Alber (Center for Biological Imaging) at University of Pittsburgh, PA for help with tracking of vacuolar movement in time lapse images. This work was supported by NIH Grant 5R01GM104412 to M.T. and C.W.L. STED imaging instrument used for this work at CBI core facility was supported through NIH Grant S10 OD021540.

## Author contributions

C.W.L., H.Y., M.S., and M.T. initiated the project concept. H.Y. and M.S. generated plasmid constructs. M.S. and M.T. generated zebrafish *myo1d* mutants and analyzed phenotypes. M.S., M.J.C., J.D.A., and M.T. performed morpholino experiments. M.S., T. F., M. Sun, and D.B.S. performed confocal, and TEM imaging and analysis. M.S., M.C., and S.C.W. performed super-resolution 4D STED and KV flow imaging and analysis. M. S. and M.T. analyzed all the data, prepared figures, and drafted the manuscript. M.T., J.D. A., and C.W.L. edited the manuscript. All the authors have read the manuscript.

## Additional information

**Competing interests:** The authors declare no competing interests.

