## [Peer Review File · Nature Communications]

Reviewers' comments:

Reviewer #1 (Remarks to the Author):

An unconventional myosin, myosin 1d regulates Kupffer's vesicle morphogenesis and laterality

The authors have demonstrated that myosin 1d is essential for establishing left-right symmetry in zebrafish. The phenotype they observe with the *myo1d* mutant, suggests that this motor plays a role in lumen formation and laterality during zebrafish development. This is a very interesting observation and the work presented is good quality.

Major points:

My biggest question is why specifically did the authors choose to substitute myosin 1d with 1c? Given the similarity of these two proteins is it really surprising that the phenotype can be rescued with *Myo1c*? I think it would be very useful to have a comparison of the two proteins in the paper and some speculation as to why this protein can compensate for the loss of *myo1d*. Some numbers of percentage homology as well as a diagram of domain structure, perhaps in the supplementary material, would also be useful.

The paper shows that *myo1d* is important in establishing left-right symmetry. Through the intracellular transport of vacuolar they show that class-I myosins can regulate cell shape and lumen formation in zebrafish KV, and they suggest that this is an evolutionary conserved mechanism of myosin-I. This is quite a bold statement and it would be very nice to see some experiments using another myosin-I or even a different myosin class to show what the general effect of overexpression of myosin has on the lumen formation.

Minor points:

Figure 1 shows that the size and circularity of the lumen is altered with the *Myo1d* MZ mutants. The representative pictures seem to show this, but the range in figure 1 b is very large. The statistics show that this is significant, but I would like to see this sample size increased for both the wildtype and *myo1d* MZ as I do not find this very convincing as it stands. Preferably the authors would quantify the circularity as well, rather than just showing a couple of example pictures.

The paper concludes with experiments centred around the CFTR pathways and the fact that *myo1d* and CFTR are two independent pathways. The authors investigated CFTR as it is localised to the KV apical membrane, and is involved in water transport into the epithelial lumen. This section, although interesting, seems like it was added on at the end and should be connected better to the rest of the paper.

Reviewer #2 (Remarks to the Author):

This manuscript explores the role of the actin-based molecular motor myosin 1D in establishing vertebrate left-right asymmetry. This is a fundamental question, since there

has been extensive discussion of whether the mechanism by which bilateral symmetry is broken is conserved from invertebrates to vertebrates. The authors convincingly show that MZ myo1d mutant zebrafish have an abnormal Kupffer's vesicle. The mutant zebrafish and KV imaging data are very convincing. They also present data supporting a role for myo1D in establishing vertebrate left-right asymmetry by showing abnormal (albeit at a very low frequency) heart looping direction in the MZ myo1d mutant fish. Although these are potentially very important findings, the proposed mechanism linking myo1d and abnormal KV and left-right development is not very well supported in the manuscript, significantly lowering the impact of the observations.

Major comments:

1. The analysis of cilia in the mutant KV is incomplete. While I agree that cilia number/cell and cilia length are not affected, there is no data to indicate that the cilia are motile, and there is no data whether the cilia have intact sensing function. Since in mouse as few as two motile cilia can provide adequate information for LR patterning, it is premature to conclude that the effect in the myo1d mutants is entirely due to a reduced cell and cilia number, and that there is no effect on ciliogenesis.
2. A direct readout of KV function is asymmetric degradation of cer12 mRNA in cells surrounding the KV. Analysis of cer12 expression in mutant embryos is essential data in this manuscript, as the data shown in the manuscript does not rule out that the low-frequency heart looping defect is due to abnormal signaling in the LPM, or abnormal competency of the heart to form a heart loop.
3. The hypothesis that myo1d is required for vacuolar trafficking and fluid transport in the KV is interesting, but not clearly supported by the data presented. The changes in vacuole movement could be entirely correlative, and not causative.
4. It is unclear how the negative data on CFTR contributes to the conclusions. It would be much more informative if data were presented showing an additive effect between myo1d-driven fluid transport and CFTR-driven fluid transport.

Minor comments:

1. Multiple grammatical errors throughout the manuscript
2. Figure 1: the preponderance of bilateral spaw expression in the mutant suggests a midline defect; was the midline intact?

Reviewer #3 (Remarks to the Author):

MyoD1 has been known to be essential for establishing LR asymmetry in *Drosophila*. In this paper, the authors have studied a role of MyoD1 in zebrafish LR asymmetry. Zebrafish MyoD1 mutants (both maternal&zygotic mutants and MO) displayed laterality defects, and the size of KV (LR organizer of zebrafish) was reduced in the Myo1D mutant. The authors found that directed vacuolar movement in KV epithelial cells was impaired in the absence of MyoD1. Although this is a very interesting observation, my major concern is that the causative relationship between the KV size reduction and LR defects remains unclear. The authors should address this, such as by manipulating the size of KV lumen of WT and

mutant embryos by pharmacological reagents (such as Quabain and IBMX). If LR defects and impaired vacuole movement are two independent events, then the authors should make this clear and investigate the mechanism for vacuole movement or lumen formation deeper.

Reviewer #1 (Remarks to the Author):

An unconventional myosin, myosin 1d regulates Kupffer's vesicle morphogenesis and laterality. The authors have demonstrated that myosin 1d is essential for establishing left-right symmetry in zebrafish. The phenotype they observe with the myo1d mutant, suggests that this motor plays a role in lumen formation and laterality during zebrafish development. This is a very interesting observation and the work presented is good quality.

Major points:

My biggest question is why specifically did the authors choose to substitute myosin 1d with 1c? Given the similarity of these two proteins is it really surprising that the phenotype can be rescued with Myo1c? I think it would be very useful to have a comparison of the two proteins in the paper and some speculation as to why this protein can compensate for the loss of myo1d. Some numbers of percentage homology as well as a diagram of domain structure, perhaps in the supplementary material, would also be useful.

Response: Amino acid alignment of Myosin 1d from zebrafish and rat, and Myosin 1c from mouse and Amoeba was performed using Clustal W algorithm in DNASTar, Lasergene software. We compared these to determine amino acid identity between the Myosin motor and the Tail-Homology 1 (TH1) domains. This is presented in **Supplementary Figure 5**. Although the overall amino acid identity between zebrafish Myo1d and Amoeba Myosin-1C is relatively low (35%), the main structural features such as motor (45%) and TH1 (21%) domains are conserved (**Supplementary Figure 5a**). Since the TH1 domain showed weak amino acid identity and this domain is known to interact with cellular membranes we reasoned that the 3D structure of this domain should still be similar between these species. The crystal structure of Mouse Myo1c TH1 domain was recently solved such that it was possible to model the TH1 domains from this class of proteins¹. We used the Phyre2 prediction server to generate PDB models of the TH1 domain². Remarkably, the Amoeba Myosin-1C TH1 domain is structured in a similar manner to mouse Myo1c, zebrafish Myo1d and Rat Myo1d TH1 domains (see **Supplementary Figure 5b**). Thus, we reasoned that it was possible to rescue lumen size of myo1d MZ mutants as we were overexpressing highly related proteins in terms of their functional domains.

The paper shows that myo1d is important in establishing left-right symmetry. Through the intracellular transport of vacuolar they show that class-I myosins can regulate cell shape and lumen formation in zebrafish KV, and they suggest that this is an evolutionary conserved mechanism of myosin-I. This is quite a bold statement and it would be very nice to see some experiments using another myosin-I or even a different

myosin class to show what the general effect of overexpression of myosin has on the lumen formation.

Response: A previous report on the structure of Myo1c from mouse showed that the class I myosins have highly conserved structured domains. Each of the class I myosins have an N-terminal Myosin motor, calmodulin IQ domains, a Tail homology domain and in some instances SH3 domains. In zebrafish there are 8 Myosin 1 family members and their function have been loosely studied. For example, gene knockdown studies reveal that *myo1b* plays a role in prechordal plate progenitor cell migration during early development³. *myo1ca*, *myo1cb*⁴ and *myo1e*⁵ are required for glomerular development and function. In this study we report that *myo1d* is required for KV lumen expansion. These studies suggest that the Class I Myosins serve tissue specific functions likely based on their protein/gene expression profile. In the case of Myo1d, although we observed only KV defects, *myo1d* is ubiquitously expressed, suggesting that its function in other cells are either not critical or that other myosins may compensate for the loss of Myo1d. We have rescued *myo1d* MZ mutants with rat *Myo1d* and amoeba *myosin-1C*. We predict that overexpression of other class 1 myosins would also rescue the KV size. Looking at other myosins such as Myosin 5 class, these proteins do have the myosin motor domains, but lack the TH1 domains. In place of the TH1 domains, are coiled-coil domains. We reasoned that overexpression of Myosin 5 class would not rescue *myo1d* mutants with the conserved TH1 domain found in all Class I myosins. Another factor that prevented us from testing this is that Myosin 5 proteins are over 1800 amino acids in length and the limitations for in vitro transcription and expression in zebrafish is about 1000 amino acids. We tried to find another myosin that was not Class I type and smaller than 1100 amino acids, but we were unable to find such genes. Thus, we are unable to address this comment as it is beyond the technical scope of what can be done with this KV lumen rescue assay.

Minor points:

Figure 1 shows that the size and circularity of the lumen is altered with the Myo1d MZ mutants. The representative pictures seem to show this, but the range in figure 1 b is very large. The statistics show that this is significant, but I would like to see this sample size increased for both the wildtype and *myo1d* MZ as I do not find this very convincing as it stands. Preferably the authors would quantify the circularity as well, rather than just showing a couple of example pictures.

Response: As per the comments, we have repeated the experiment and increased the sample size to see if it has any effect on statistical significance and conclusion derived from the experiment in the previous version of the manuscript. Frequency of occurrence of KV with small lumen consistently increased in the mutants over controls (please see **Figure 1d**). We are thankful to the reviewer for suggesting to measure KV circularity or isoperimetric quotient, a function of perimeter and area. It is indeed a critical information that helps us to explain the role of *myo1d* in regulating laterality. Please find the data on circularity added to the manuscript **Figures 1e, 4k, 5h**. In order to substantiate our

results, we also have additional pictures of dysmorphic lumen found in *myo1d* MZ mutants over controls (**Supplementary Figure 1d**).

The paper concludes with experiments centred around the CFTR pathways and the fact that *myo1d* and CFTR are two independent pathways. The authors investigated CFTR as it is localised to the KV apical membrane, and is involved in water transport into the epithelial lumen. This section, although interesting, seems like it was added on at the end and should be connected better to the rest of the paper.

Response: Cystic fibrosis transmembrane conductance regulator (CFTR) is localized in KV apical membranes and was shown to regulate water transport into the epithelial lumen⁶. Consistently, CFTR mutant zebrafish exhibited impaired KV lumen formation⁷. Also, pharmacological treatment with Forskolin and IBMX (FIBMX) that activate CFTR, expanded KV lumen size⁸. A recent report showed that Myo1a, another unconventional myosin is required for CFTR brush border membrane trafficking and ion transport in the mouse intestine⁹. Thus, Myo1d could also be involved in CFTR trafficking to the membrane that is a separate mechanism of vacuole transport. We have included the rationale in the text that lead to our experiments to show that CFTR localization in KV apical surface is not regulated through *myo1d* in zebrafish.

Reviewer #2 (Remarks to the Author):

This manuscript explores the role of the actin-based molecular motor myosin 1D in establishing vertebrate left-right asymmetry. This is a fundamental question, since there has been extensive discussion of whether the mechanism by which bilateral symmetry is broken is conserved from invertebrates to vertebrates. The authors convincingly show that MZ *myo1d* mutant zebrafish have an abnormal Kupffer's vesicle. The mutant zebrafish and KV imaging data are very convincing. They also present data supporting a role for *myo1D* in establishing vertebrate left-right asymmetry by showing abnormal (albeit at a very low frequency) heart looping direction in the MZ *myo1d* mutant fish. Although these are potentially very important findings, the proposed mechanism linking *myo1d* and abnormal KV and left-right development is not very well supported in the manuscript, significantly lowering the impact of the observations.

Major comments:

Response: We are indeed excited with the positive comments from reviewer #2 regarding this work. The major question we addressed in the revised submission is to clarify how *myo1d* regulates laterality through formation of a proper KV size and shape.

1. The analysis of cilia in the mutant KV is incomplete. While I agree that cilia number/cell and cilia length are not affected, there is no data to indicate that the cilia are motile, and there is no data whether the cilia have intact sensing function. Since in mouse as few as two motile cilia can provide adequate information for LR patterning, it is premature to conclude that the effect in the *myo1d* mutants is entirely due to a reduced cell and cilia number, and that there is no effect on ciliogenesis.

Response: This is a very critical comment and we truly appreciate the suggestion. We have performed KV flow analysis by injecting fluorescent microbeads into wildtype and *myo1d* MZ mutant KVs. We have included new data showing that *myo1d* MZ mutants have erratic flow (please see **Fig. 2e & f**), suggesting that cilia are motile and functional in the context of a smaller and dysmorphic KV lumen. This erratic flow could be a consequence of dysmorphic lumen formation and not due to ciliogenic defects such as length and number. Although cilia are motile, we have not quantified cilia motion in detail to rule out cilia motion defects. This could be a relevant phenotype, however we believe it will be difficult to separate from the disruption of proper lumen formation.

2. A direct readout of KV function is asymmetric degradation of *cerl2* mRNA in cells surrounding the KV. Analysis of *cerl2* expression in mutant embryos is essential data in this manuscript, as the data shown in the manuscript does not rule out that the low-frequency heart looping defect is due to abnormal signaling in the LPM, or abnormal competency of the heart to form a heart loop.

Response: We like to thank the reviewer for this critical suggestion. During KV morphogenesis, one of the major transcriptional responses is asymmetric expression of the TGF β antagonist, *dand5* (also known as *Charon*, and *cerl2*)¹⁰. In *myo1d* MZ mutants, a majority showed diffuse or abnormal *dand5* expression that is indicative of a failure of proper KV morphogenesis (**Fig. 2a & b**). Consequently, *spaw* was found to have higher frequency of bilateral expression and will translate into cardiac and gut looping defects (**Fig. 2c**).

3. The hypothesis that *myo1d* is required for vacuolar trafficking and fluid transport in the KV is interesting, but not clearly supported by the data presented. The changes in vacuole movement could be entirely correlative, and not causative.

Response: Mutation of *myo1d* in zebrafish lead to the appearance of increased frequency of larger vacuoles during KV lumen development (**Fig 3**). Further analysis revealed that speed of vacuole movement is reduced in the mutant compared to wildtype controls at comparable developmental phase. From this data, it appears to be correlative that vacuole appearance is on the same developmental phase as that of wildtype controls. On the other hand, in order to test if it is causative relationship, we injected rat *myo1d* mRNA in zebrafish *myo1d* MZ mutants and quantified vacuole volume when KV is fully formed (about 8 S stage) (**Fig. 4i & j**). We observed a marked decrease in vacuole volume in the KV cells of *myo1d* mutants injected with Rat *Myo1d* mRNA compared to the non-injected *myo1d* mutants. This result has established that *Myo1d* likely mediated vacuole delivery as the cause of lumen expansion.

4. It is unclear how the negative data on CFTR contributes to the conclusions. It would be much more informative if data were presented showing an additive effect between *myo1d*-driven fluid transport and CFTR-driven fluid transport.

Response: In our experiment, FIBMX treatment in *myo1d* MZ embryos restored KV lumen size, but not shape, implicating CFTR is still functional in the absence of *Myo1d*

(**Fig. 5c-e**), and that *myo1d* is not required for CFTR apical membrane localization. One study had reported that Myo1a is required for CFTR localization to the brush border membrane of mouse intestinal epithelium⁹. We wanted to rule out the potential for *myo1d* serving a similar function in zebrafish KV. Our experiments show that Myo1d mutation does not alter CFTR localization to the KV apical membrane as we were able to increase lumen size in the *myo1d* mutant with FIBMX treatment. However, FIBMX treatment failed to regain circular shape (**Fig. 5d, g**) suggesting that Myo1d function is also critical for regulating circularity, presumably by delivering plasma membrane to the apical surface. We posit that having a circular shape is vital to create a unidirectional flow in KV and that delivery of vacuole membrane and fluid by Myo1d is essential for both KV shape and size. The ion gradient driving CFTR dependent fluid secretion is closely associated with the function of the Na⁺/K⁺-ATPase channel. So we treated with ouabain, a potent specific inhibitor Na⁺/K⁺-ATPase to check the additive effect of Myo1d and CFTR in KV lumen formation⁷. In control embryos, ouabain treatment decreased lumen volume similar to previous reports (**Fig 5c-e**)^{7,8}, validating our pharmacological treatment protocol settings. In *myo1d* MZ mutants, ouabain treatment did show a trend to smaller lumen size beyond untreated *myo1d* MZ mutants (**Fig. 5c, d& e**). However, this decrease in size did not translate to increased laterality defects (**Fig. 5f**). These experiments suggest that a minimum threshold KV lumen size and shape is critical for proper left-right patterning and *myo1d* is an essential component for KV morphogenesis.

Minor comments:

1. Multiple grammatical errors throughout the manuscript
2. Figure 1: the preponderance of bilateral spaw expression in the mutant suggests a midline defect; was the midline intact?

Response:

1. We thank the reviewer for pointing out grammatical errors. We have addressed grammatical errors in the revised manuscript.
2. We have included the data showing that midline is intact in *myo1d* MZ mutants using *tbxta* (recently changed from *ntl*) that marks the notochord, and *myo1*, a somite and adaxial cell marker (see **Supplementary Figure 4b**).

Reviewer #3 (Remarks to the Author):

MyoD1 has been known to be essential for establishing LR asymmetry in *Drosophila*. In this paper, the authors have studied a role of MyoD1 in zebrafish LR asymmetry. Zebrafish MyoD1 mutants (both maternal& zygotic mutants and MO) displayed laterality defects, and the size of KV (LR organizer of zebrafish) was reduced in the Myo1D mutant. The authors found that directed vacuolar movement in KV epithelial cells was impaired in the absence of Myo1. Although this is a very interesting observation, my

major concern is that the causative relationship between the KV size reduction and LR defects remains unclear. The authors should address this, such as by manipulating the size of KV lumen of WT and mutant embryos by pharmacological reagents (such as Ouabain and IBMX). If LR defects and impaired vacuole movement are two independent events, then the authors should make this clear and investigate the mechanism for vacuole movement or lumen formation deeper.

We thank Reviewer#3 for their positive response to our findings. There is a debate that the observed background laterality defects in wildtype strains is a result of variabilities in KV size. A recent study by the Amack group explored this possibility leading to the hypothesis that KV lumen size does influence laterality¹¹. We have addressed the concept of lumen size further with new experiments using small molecules to alter KV lumen size in wildtype and *myo1d* mutant embryos as suggested by the reviewer. Expanding or decreasing lumen size with FIBMX and ouabain, respectively did increase cardiac looping defects in WT embryos (see **Fig. 5c, d, & e**). However, in *myo1d* mutants, although FIBMX did restore lumen size, it did not rescue cardiac looping defects (**Fig. 5f**). Similarly, treatment with ouabain in wildtype embryos decreased lumen size and increased cardiac looping defects (**Fig. 5c, d, & e**). In *myo1d* mutants, ouabain did not change the incidence of heart looping defects (**Fig. 5f**). One possible explanation as to why we were not able to rescue laterality based on lumen size is that in these experiments, we invariably failed to restore proper spherical shape of the lumen. We hypothesize that the vacuole trafficking to the apical surface is also important to supply plasma membrane such that the KV can expand and maintain a spherical structure. Pharmacological activation of CFTR only changes volume but does not increase membrane in the absence of *myo1d*. This is a contributing factor for forming spherical shape during KV lumen expansion and suggests that both CFTR and *Myo1d* are required for proper KV formation in a developmental restricted time period for specifying laterality.

References

- 1 Lu, Q., Li, J. C., Ye, F. & Zhang, M. J. Structure of myosin-1c tail bound to calmodulin provides insights into calcium-mediated conformational coupling. *Nat Struct Mol Biol* **22**, 81-88, doi:10.1038/nsmb.2923 (2015).
- 2 Kelley, L. A., Mezulis, S., Yates, C. M., Wass, M. N. & Sternberg, M. J. E. The Phyre2 web portal for protein modeling, prediction and analysis. *Nat Protoc* **10**, 845-858, doi:10.1038/nprot.2015.053 (2015).
- 3 Diz-Munoz, A. *et al.* Control of directed cell migration in vivo by membrane-to-cortex attachment. *PLoS Biol* **8**, e1000544, doi:10.1371/journal.pbio.1000544 (2010).
- 4 Arif, E. *et al.* Myo1c is an unconventional myosin required for zebrafish glomerular development. *Kidney Int* **84**, 1154-1165, doi:10.1038/ki.2013.201 (2013).
- 5 Mao, J. H. *et al.* Myo1e Impairment Results in Actin Reorganization, Podocyte Dysfunction, and Proteinuria in Zebrafish and Cultured Podocytes. *Plos One* **8**, doi:ARTN e7275010.1371/journal.pone.0072750 (2013).
- 6 Riordan, J. R. CFTR function and prospects for therapy. *Annu Rev Biochem* **77**, 701-726, doi:10.1146/annurev.biochem.75.103004.142532 (2008).

- 7 Navis, A., Marjoram, L. & Bagnat, M. Cftr controls lumen expansion and function of Kupffer's vesicle in zebrafish. *Development* **140**, 1703-1712, doi:10.1242/dev.091819 (2013).
- 8 Roxo-Rosa, M., Jacinto, R., Sampaio, P. & Lopes, S. S. The zebrafish Kupffer's vesicle as a model system for the molecular mechanisms by which the lack of Polycystin-2 leads to stimulation of CFTR. *Biol Open* **4**, 1356-1366, doi:10.1242/bio.014076 (2015).
- 9 Kravtsov, D. V. *et al.* Myosin Ia is Required for CFTR Brush Border Membrane Trafficking and Ion Transport in the Mouse Small Intestine. *Traffic* **13**, 1072-1082, doi:10.1111/j.1600-0854.2012.01368.x (2012).
- 10 Hashimoto, H. *et al.* The Cerberus/Dan-family protein Charon is a negative regulator of Nodal signaling during left-right patterning in zebrafish. *Development* **131**, 1741-1753, doi:10.1242/dev.01070 (2004).
- 11 Gokey, J. J., Ji, Y., Tay, H. G., Litts, B. & Amack, J. D. Kupffer's vesicle size threshold for robust left-right patterning of the zebrafish embryo. *Dev Dyn* **245**, 22-33, doi:10.1002/dvdy.24355 (2016).

Reviewers' comments:

Reviewer #1 (Remarks to the Author):

The authors have satisfactorily addressed all my comments, I recommend the paper for publication.

Reviewer #3 (Remarks to the Author):

This paper is highly related to Juan et al. on bioRxiv (<https://www.biorxiv.org/content/early/2018/02/16/267146>), which is also listed as under consideration at Nature Communications. Also, one more related paper (Tingler et al) was recently published in Current Biology during revision of this paper. Although conclusion of three papers are the same in that Myo1d is required for L-R asymmetry in zebrafish or frog, the exact role of Myo1d in L-R asymmetry is quite different.

1) This paper suggests that the reduction of the LR organizer (KV) is the primary reason for the laterality defects of Myo1d mutant, whereas the Juan et al. paper provides compelling evidence that the reduction of KV is NOT the primary reason. In particular, those two papers contain very similar experiment but the results are different: treatment of WT zebrafish embryos with 1 microM Ouabain reduced the KV size and increased L-R defects in this paper, while similar treatment of WT zebrafish embryos with 5 microM Ouabain reduced KV size but no significant effects on LR asymmetry. Other data such as the effects of FIBMX on WT and Myo1d mutant embryos are similar in both papers.

2) The Juan et al paper shows abnormal positioning of the basal body in the Myo1d mutant and genetic interaction between Myo1d and Vangl2, and they suggest that Myo1d functions within the PCP pathway. The Tingler et al paper also suggests a similar conclusion, although their experimental evidence is much weaker: frog embryos treated with Myo1d-MO showed neural tube closure defects, which are indicative of PCP defects. This has not been addressed by the Saydmohammed et al paper.

When I evaluate this paper separately from two other papers, the data may be solid, but the paper does not have a big impact on the LR field. If the size of KV is severely reduced, this would naturally induce LR defects (no surprise). If Myosin1d regulates the transport of vacuoles, as the authors claim, it is interesting but I would like to see its more precise role in vacuole transporting. In any case, I cannot recommend this paper with enthusiasm.

Additional comments from reviewer #3 on the response to reviewer #2's concerns:

<Major comments>

#1. As requested by the reviewer, the authors have tracked fluorescent microbeads in the

KV lumen, and found "erratic" fluid flow in the myo1d MZ mutant. At least, it is now clear that cilia in KV are motile. They simply mention that the flow is "erratic" (line 166), but "erratic" needs to be described more precisely. Also, the way to show trajectories can be improved.

#2. As requested by the reviewer, they have examined Cerl2/dand5 mRNA at the KV. In the majority of the mutant embryos examined, Cerl2/dand5 mRNA was "abnormal or diffused". This is not a scientific description of Cerl2/dand5 mRNA: they should analyze the data in more qualitative and quantitative manner. Critical issue here is whether it is L-R asymmetric or not, and whether the level on each side is changed or not. They did the right experiments but did not analyze properly. This is obvious when compared to the another paper (Figure 2 of Juan et al), where they carefully followed the mRNA at different stages and showed that L-R asymmetry of Cerl2/dand5 mRNA is disrupted in the mutant.

#3. Whether changes in vacuole movement is correlative or causative to mutant LR defects. This is a difficult issue to address, but rescue of vacuole volume (Fig. 4j), and the circularity (Fig. 4k) by rat Myo1d would not be a proof. If vacuole transport and LR defects are simultaneously rescued by a treatment not directly involving myo1d gene, then this would be a more direct proof.

#4. This was also pointed out by reviewer #1. The authors have now explained the rationale of examining CFTR. The current data (Fig. 5) simply suggest that myo1d and CFTR are two independent pathways, but do not add supports to their conclusion.

A related issue: Two papers examined effects of FIBMX and reported similar observations: FIBMX on myo1d mutant increased the KV lumen size but did not rescue cardiac looping defects. Based on these observations, Yuan et al suggest that the size reduction of KV lumen is NOT the reason for L-R defects. On the other hand, this paper claims that the shape of KV (the circularity: i.e. roundness) is not rescued by FIBMX (Fig. 5f), which is the reason why cardiac looping defects were not rescued. They say that the circularity was calculated by measuring the area and perimeter of KV lumen, but I could not find the reference (Prog. Earth Planet Sci. 2016). Is it right to calculate the circularity (roundness) from area and perimeter? Circularity value must be independent of the size of a shape. I would like to know how exactly they calculated the circularity: for example, what is the circularity value for the KV lumen of WT embryo, and how it was estimated.

<Minor comments>

They have responded satisfactorily.

Reviewers' comments:

Reviewer #1 (Remarks to the Author):

The authors have satisfactorily addressed all my comments, I recommend the paper for publication.

Reviewer #3 (Remarks to the Author):

This paper is highly related to Juan et al. on bioRxiv (<https://www.biorxiv.org/content/early/2018/02/16/267146>), which is also listed as under consideration at Nature Communications. Also, one more related paper (Tingler et al) was recently published in Current Biology during revision of this paper. Although conclusion of three papers are the same in that Myo1d is required for L-R asymmetry in zebrafish or frog, the exact role of Myo1d in L-R asymmetry is quite different.

During the review and revision of this study there have been two studies that describe the role of Myo1d in left-right patterning. We are happy to see that the zebrafish mutant generated by Juan et al. also come to similar findings with smaller and dysmorphic KVs. However, their subsequent studies focused on cilia orientation as the primary defect that drive the laterality phenotypes. In our studies we addressed the role of Myo1d in KV lumen morphogenesis. Although these may appear as two separate defects, we do not feel that the two conclusions are incongruent. First, the KV morphology defect would likely affect cilia orientation, therefore the fluid flow mechanism driven by the failure of proper cilia orientation is anticipated. Juan et al. and our studies have demonstrated this. Second, our studies address the reason why KV morphology is affected and how myosin I protein is utilized to traffic vacuoles to the apical surface and expand the KV lumen. This is a new finding and adds to the complexity of fluid-filling mechanism of the KV organ. The reason for this complexity is probably linked to the short developmental time frame for the KV to function as a transient structure. This 3 hour time period is short and multiple mechanism are required for proper KV to form.

1) This paper suggests that the reduction of the LR organizer (KV) is the primary reason for the laterality defects of Myo1d mutant, whereas the Juan et al. paper provides compelling evidence that the reduction of KV is NOT the primary reason. In particular, those two papers contain very similar experiment but the results are different: treatment of WT zebrafish embryos with 1 microM Ouabain reduced the KV size and increased L-R defects in this paper, while similar treatment of WT zebrafish embryos with 5 microM Ouabain reduced KV size but no significant effects on LR asymmetry. Other data such as the effects of FIBMX on WT and Myo1d mutant embryos are similar in both papers.

The first phenotype we observed in the *myo1d* MZ mutant embryos were smaller KVs. We have carefully documented the decreased and dysmorphic lumen size as a primary reason for the increased laterality defects observed (please see Fig. 1c-e). Given with what has been previously associated with small KV lumen size and laterality as described by: Gokey et. al Dev Dyn. 245:22-33 (2016), Compagnon et. al. Dev Cell 31:774-783 (2014), Kim et. al. PLoS One 12(8):e0182047 (2017), and Navis et al. Dev. 140:1703-1712 (2013), we further address the role of *myo1d* in regulating lumen expansion. In all these studies, small lumen size was associated with left-right patterning defects. Moreover, some of these studies used Ouabain as a chemical inhibitor of Na⁺/K⁺ ATPase to address the role of ion channels in efflux of water as a mechanism of lumen expansion. To date all have shown reduced lumen size with Ouabain treatment and the associated increase in laterality defects. We have repeated the studies with Ouabain treatment with different doses and noted laterality defects. We are not sure why the study now published by Juan et al. did not see a similar effect with Ouabain on WT embryos. It has been noted that short treatment periods with Ouabain at 4-6S did not alter laterality or lumen size so that could one reason there is a discrepancy between our studies. We are at a loss as to how to address this other than repeating these experiments, which we have done (Supplementary Figure 6) and see similar results as the other studies.

2) The Juan et al paper shows abnormal positioning of the basal body in the Myo1d mutant and genetic interaction between Myo1d and Vangl2, and they suggest that Myo1d functions within the PCP pathway. The Tingler et al paper also suggests a similar conclusion, although their experimental evidence is much weaker: frog embryos treated with Myo1d-MO showed neural tube closure defects, which are indicative of PCP defects. This has not been addressed by the Saydmohammed et al paper.

The role of *myo1d* in PCP pathway is intriguing and clearly presented by Tingler et. al. in *Xenopus* MO studies. In our hands neither the MO or the *myo1d* mutant presented with PCP phenotypes. We have addressed this by observing normal gastrulation of *myo1d* MZ mutant embryos as there was no delay in development during gastrulation, a standard phenotype associated with PCP defects. Supplementary Figure 3 show normal expression and progression of Dorsal Forerunner Cells (DFC) in *myo1d* MZ mutant and expression of *tbxta* (*no-tail a*) was normal and did not

show typical gastrulation defects such as wider/expanded notochords at gastrulation stages. We believe we have addressed this issue of PCP defects in the supplementary figure 3.

When I evaluate this paper separately from two other papers, the data may be solid, but the paper does not have a big impact on the LR field. If the size of KV is severely reduced, this would naturally induce LR defects (no surprise). If Myosin1d regulates the transport of vacuoles, as the authors claim, it is interesting but I would like to see its more precise role in vacuole transporting. In any case, I cannot recommend this paper with enthusiasm.

In this study, we present a new mechanism of how the KV lumen expands during a 3-hour time window. Other studies have presented evidence of ion channels and Cfr as primary effectors of KV lumen. However, in this study we identified a role for Myo1d in regulating trafficking of fluid-filled vacuoles as an independent mechanism in this process. It is remarkable that the function of Myosin I is conserved from amoeba to vertebrates and that this process controls water regulation in a single cell organism can be applied to regulating transient fluid filled structures critical for dictating higher order laterality in a multi-cellular organism. Thus, we believe that this study does offer new insights into how fluid filled structures form and adds another level of complexity with ion channels.

To address the implication that Myo1d regulates vacuole transport, we have generated a myosin construct without the TH1 domain. Previous studies have implicated the TH1 domain can interact with membrane structures. We injected the rat Myo1d without a TH1 domain and showed that this failed to rescue KV lumen area as full-length Myo1d did. These new experiments support the model that Myo1d functions to transport vacuoles with the TH1 domain serving as the cargo binding domain.

Additional comments from reviewer #3 on the response to reviewer #2's concerns:

<Major comments>

#1. As requested by the reviewer, the authors have tracked fluorescent microbeads in the KV lumen, and found "erratic" fluid flow in the *myo1d* MZ mutant. At least, it is now clear that cilia in KV are motile. They simply mention that the flow is "erratic" (line 166), but "erratic" needs to be described more precisely. Also, the way to show trajectories can be improved.

We have calculated bead tracks in the KV to assess the KV flow speed and direction of flow. In our study, we observed unidirectional flow in the WT embryos at 8S stage (Figure 2 and supplementary Movie 3). However, in the mutant we observed failure of unidirectional flow. As per the reviewer's suggestion, we have rephrased "erratic" flow as "failure of unidirectional flow" in text.

The bead track speed were expressed as $\mu\text{m/s}$ similar to previous reports (Wang et al., *J Vis Exp.* 2013; (73): 50038.). Angular velocity is another way of representing the KV flow (as reported in Juan et. al.), which is a measure of linear velocity multiplied by the radius. Since the KV shape are often dysmorphic and not circular in *myo1d* MZ mutants, we reasoned that this may not be a reliable measure for KV flow. Trajectories of beads have been improved and shown in Figure 2e.

#2. As requested by the reviewer, they have examined *Cerl2/dand5* mRNA at the KV. In the majority of the mutant embryos examined, *Cerl2/dand5* mRNA was "abnormal or diffused". This is not a scientific description of *Cerl2/dand5* mRNA: they should analyze the data in more qualitative and quantitative manner. Critical issue here is whether it is L-R asymmetric or not, and whether the level on each side is changed or not. They did the right experiments but did not analyzed properly. This is obvious when compared to the another paper (Figure 2 of Juan et al), where they carefully followed the mRNA at different stages and showed that L-R asymmetry of *Cerl2/dand5* mRNA is disrupted in the mutant.

We agree with the reviewers' assertion that the expression of *dand5* was not properly quantified. We have repeated *in situ* to detect *dand5* expression in WT and *myo1d* MZ mutant embryos and quantified as previously described by Superina et. al. Consistent with Juans paper, we found that *dand5* mRNA right to left to right ratio was reduced in the mutant at 8S stage (See new figure 2a, b).

#3. Whether changes in vacuole movement is correlative or causative to mutant LR defects. This is a difficult issue to address, but rescue of vacuole volume (Fig. 4j), and the circularity (Fig. 4k) by rat Myo1d would not be a proof. If vacuole transport and LR defects are simultaneously rescued by a treatment not directly involving *myo1d* gene, then this would be a more direct proof.

The experiments suggested by the reviewer is challenging to perform. In order to image the vacuole transport, quantify the change and then address the issue of laterality is nearly impossible to do. For 4D STED imaging, the embryos are dechorionated and placed in low melt agarose to stabilize for super resolution imaging. After imaging we would have to correlate vacuole movement, with lumen area and then address the issue of laterality. Typically, we are only able to image 3-5 embryos per day as the time window of KV morphogenesis is too short in order to process more samples. In addition, it is challenging to extract embryos from low-melt agarose and allow them to develop until 48 hpf in order to monitor laterality. We have tried to correlate lumen size with FIBMX treatment in *myo1d* MZ mutants and quantify laterality defects. In these experiments, although lumen size was expanded with FIBMX treatment, we were unable to restore laterality (see Figure 5e). A reason for this is that the exact size and circularity is not possible with these treatments. Another correlation is our new experiment showing the TH1 domain in Myo1d is critical for restoring lumen size suggesting that this domain interacts with vacuole membranes for transport.

#4. This was also pointed out by reviewer #1. The authors have now explained the rationale of examining CFTR. The current data (Fig. 5) simply suggest that *myo1d* and CFTR are two independent pathways, but do not add supports to their conclusion.

There have been numerous studies on the role of ion channels and Cfr in regulating lumen size. Since Myosin 1a has been implicated in trafficking Cfr to apical membranes, we reasoned that this could be an explanation for the decreased KV lumen size as observed in *myo1d* MZ mutants. This would invalidate our hypothesis that Myo1d regulates KV lumen size by vacuole trafficking in epithelial cells. While the reviewer is correct that these experiments do not fully support the conclusion that Myo1d regulates vacuole trafficking, it does eliminate the requirement for *myo1d* in Cfr trafficking. Future studies could address the genetic interaction between *cfr* and *myo1d*.

A related issue: Two papers examined effects of FIBMX and reported similar observations: FIBMX on *myo1d* mutant increased the KV lumen size but did not rescue cardiac looping defects. Based on these observations, Yuan et al suggest that the size reduction of KV lumen is NOT the reason for L-R defects. On the other hand, this paper claims that the shape of KV (the circularity: i.e. roundness) is not rescued by FIBMX (Fig. 5f), which is the reason why cardiac looping defects were not rescued. They say that the circularity was calculated by measuring the area and perimeter of KV lumen, but I could not find the reference (Prog. Earth Planet Sci. 2016). Is it right to calculate the circularity (roundness) from area and perimeter? Circularity value must be independent of the size of a shape. I would like to know how exactly they calculated the circularity: for example, what is the circularity value for the KV lumen of WT embryo, and how it was estimated.

In previous revision stage, reviewer#2 suggested that we look into circularity as another parameter that Myo1d regulates during KV lumen expansion and laterality. As per this suggestion, we incorporated the data calculating circularity from the largest surface area of KV lumen in the last revision. As reviewer #3 suggested, we have now included detailed information and the formula used to calculate circularity in the methods section to state: "For measuring circularity, largest Z stack of ZO1 stained images were used to measure area and perimeter and calculated the circularity values as described previously by Pantic, I. et al. *Chromatin Fractal Organization, Textural Patterns, and Circularity of Nuclear Envelope in Adrenal Zona Fasciculata Cells. Microsc Microanal* **22**, 1120-1127 (2016), using the formula: $f = \frac{4\pi A}{p^2}$." We have included the reference we used to calculate circularity.

We have now performed new experiments to carefully map the circularity in WT embryos during early somitogenesis stages. In the case of wild type embryos, circularity is maintained during lumen expansion phase between 3 and 8 S stage (Supplementary Fig.1e). However, in the mutants, dysmorphic lumen and failure of circularity was observed. In this revision we have expanded the discussion section and added a deeper description of these findings in light of the newly published work from Tingler et al and Juan et al. We have highlighted in yellow where we have made major changes to the text.

<Minor comments>

They have responded satisfactorily.

REVIEWERS' COMMENTS:

Reviewer #3 (Remarks to the Author):

In response to my concerns and those from reviewer #2, the authors have provided additional data that support their conclusion. Perhaps, this is the best one can do. I have no further comment, and the paper can be accepted.

Although discrepancies still remain between this paper and two other papers, it is difficult to judge which is correct. We must wait for follow-up studies in future.

REVIEWERS' COMMENTS:

Reviewer #3 (Remarks to the Author):

In response to my concerns and those from reviewer #2, the authors have provided additional data that support their conclusion. Perhaps, this is the best one can do. I have no further comment, and the paper can be accepted.

Although discrepancies still remain between this paper and two other papers, it is difficult to judge which is correct. We must wait for follow-up studies in future.

Response:

Reviewer #3 is satisfied with our additional data to support our conclusions. We thank the reviewer for taking the time and effort to review our manuscript.